# A Comparison of the Australian Dietary Guidelines to the NOVA Classification System in Classifying Foods to Predict Energy Intakes and Body Mass Index

**DOI:** 10.3390/nu14193942

**Published:** 2022-09-23

**Authors:** Amanda Grech, Anna Rangan, Margaret Allman-Farinelli, Stephen J. Simpson, Tim Gill, David Raubenheimer

**Affiliations:** 1Charles Perkins Centre, University of Sydney, Sydney, NSW 2006, Australia; 2School of Life and Environmental Science, Faculty of Science, University of Sydney, Sydney, NSW 2006, Australia; 3School of Nursing, Faculty of Medicine and Health, University of Sydney, Sydney, NSW 2006, Australia

**Keywords:** obesity, ultra-processed foods, dietary guidelines, non communicable disease, protein leverage hypothesis, macronutrient intake

## Abstract

NOVA classification distinguishes foods by level of processing, with evidence suggesting that a high intake of ultra-processed foods (UPFs, NOVA category 4) leads to obesity. The Australian Dietary Guidelines, in contrast, discourage excess consumption of “discretionary foods” (DFs), defined according to their composition. Here, we (i) compare the classification of Australian foods under the two systems, (ii) evaluate their performance in predicting energy intakes and body mass index (BMI) in free-living Australians, and (iii) relate these outcomes to the protein leverage hypothesis of obesity. Secondary analysis of the Australian National Nutrition and Physical Activity Survey was conducted. Non-protein energy intake increased by 2.1 MJ (*p* < 0.001) between lowest and highest tertiles of DF intake, which was significantly higher than UPF (0.6 MJ, *p* < 0.001). This demonstrates that, for Australia, the DF classification better distinguishes foods associated with high energy intakes than does the NOVA system. BMI was positively associated with both DFs (−1. 0, *p* = 0.0001) and UPFs (−1.1, *p* = 0.0001) consumption, with no difference in strength of association. For both classifications, macronutrient and energy intakes conformed closely to the predictions of protein leverage. We account for the similarities and differences in performance of the two systems in an analysis of Australian foods.

## 1. Introduction

The proliferation of industrially formulated, affordable, palatable, aggressively marketed, and convenient foods has been linked to the rise in obesity and certain noncommunicable diseases [1]. Classification systems have thus been developed which distinguish foods according to the degree of industrial processing [2,3,4]. The NOVA system is the most applied processed food classification system in the academic literature [1]. It uses four categories of processing to classify foods. Category 1 features unprocessed and minimally processed foods, category 2 features culinary ingredients, category 3 features processed foods (combining ingredients from categories 1 and 2), and category 4 features ultra-processed foods (UPFs) [5]. UPFs are industrially produced foods that have been formulated with cosmetic food additives such as flavours, colours, and emulsifiers, with or without the addition of ingredients such as cheap oils, refined sugars and starches, and added salt [5]. Although the NOVA system is the most prominent processing-based classification system, despite accumulating evidence for the health risks associated with increased consumption of UPFs, it has not been adopted widely as the basis for dietary guidance [1]. Surprisingly, there are few direct comparisons of processing-based and alternative classification systems that are already widely used.

Most diet classification systems have attempted to identify unhealthy foods based on nutrient composition [6,7,8,9,10,11]. One example is the “discretionary foods” category in the Australian Dietary Guidelines (ADG). The ADG identifies healthy foods but contrasts with the USDA guidelines because it has a discrete category for less healthy foods [6,11]. These are defined as energy-dense, nutrient-poor foods that do not fit within one of five food group categories and are characterised by their high content of added sugars, saturated fats, added salt, and alcoholic beverages [6]. Discretionary foods are advised to be avoided to minimise the risk of noncommunicable disease and for weight maintenance, due to their high energy density and low nutrient content [6]. The systematic process adopted in the development of the ADG has been appraised to be amongst the most rigorous of 32 food-based dietary guidelines considered [12].

Fundamental to determining whether classification systems based on degree of industrial processing or nutrient content are more accurate at identifying foods that should be limited in the diet is understanding how these foods interact with appetite and metabolic biology to increase energy intakes and predispose populations to weight gain and poor health [13]. One possible mechanism is the protein leverage model, which proposes that, when faced with food environments where protein is diluted by fat and carbohydrate, the dominant human appetite for protein drives energy over consumption as an inadvertent outcome of the strong drive to meet the protein target [14,15,16]. There is a growing body of evidence to support the protein leverage model as a significant determinant of excess energy intake and obesity [16,17,18].

Two studies have directly linked UPF to the protein leverage model. An RCT conducted in a controlled environment demonstrated that UPF intake contributes to higher energy intakes and weight gain compared to a diet composed of minimally processed foods. As predicted by the protein leverage model, participants consumed foods that amounted to the same intake of protein (490 kcal) on both the minimally processed and the UPF diets, but non-protein energy increased by 509 kcal on the UPF diet [19]. Similarly, in a population study, increased UPF intake was associated with protein dilution and higher energy intakes in NHANES, whereas protein intake remained near constant [20]. However, no study has examined discretionary foods in the context of the protein leverage mechanism, nor compared the performance in predicting excess energy intake and BMI of this nutrient-based classification with processing-based classification systems.

The primary aim of the present analysis was to compare NOVA classification system with the ADG in classifying foods as healthy and unhealthy through their efficacy in predicting energy over-consumption and BMI within the Australian food system. We do so in an analysis of the National Nutrition and Physical Activity Survey, a large representative cross-sectional study of the Australian population, through (i) a comparison of the foods classified as healthy and unhealthy between the two systems, (ii) a comparison of the performance of the two systems in predicting energy intakes and body mass index, and (iii) a comparison of how discretionary food vs. UPF relate to the mechanistic model of protein leverage for predicting the relationship between dietary composition and energy intake.

## 2. Materials and Methods

### 2.1. Study Design

The data for the present analysis came from the National Nutrition and Physical Activity Survey, 2011–2012 [21]. The survey was conducted by the Australian Bureau of Statistics (ABS) using a stratified, multistage area sample of private dwellings across Australia to provide a representative sample of the population of the country. A random subsample of residents from each household were selected to participate in the survey. Eligible members of the household included one adult (aged 19 years and older) and (where applicable) one child aged 2–18 years (NNPAS). The survey was conducted between 29 May 2011 to 9 June 2012. Collection days included Monday through to Sunday. Data were collected by trained ABS interviewers, through computer-assisted personal interview (CAPI) with eligible members of the household. For the present analysis, adults aged 19 years and over were included. Full details of the survey are published elsewhere [21]. Ethics approval was granted under the Census Act 1905 by the Australian Government in accordance with relevant guidelines and regulations, and written and informed consent was obtained from all subjects and/or their legal guardian(s) [21].

### 2.2. Variables

The exposure of interest was the percentage of total energy (%E) consumed from UPF and from discretionary foods. The outcomes included energy intake from protein and non-protein sources and body mass index.

Foods were categorised by the level of processing using the NOVA food classification system [5,22]. NOVA categories distinguish four groups of foods: (1) unprocessed or minimally processed foods (including fresh, frozen, squeezed, and dried foods); (2) culinary ingredients (e.g., flour, sugar, salt and vegetable oils); (3) processed foods, which are composite foods including ingredients from both group 1 and group 2 (e.g., canned or bottled group 1 foods); (4) UPF, which are often highly concentrated in fats and sugars from group 2 and include ingredients that have been highly processed or are industrially manufactured to aid processing and not normally found in culinary preparations. Examples of such ingredients include dyes, colours, flavour enhancers, flavourings, anti-caking agents, and emulsifiers or can include extracts derived from processing group 1 ingredients such as hydrolysed proteins, hydrogenated oils, high-fructose corn syrup, and maltodextrin. Detailed information on the classification system and the application of the NOVA classification system to the NNPAS is published in detail elsewhere [5,22].

Homemade mixed dishes and composite foods and recipes that were not deemed UPF, disaggregated into their individual components, and classified into their corresponding NOVA group. The recipes for 2585 mixed dishes and composite foods have been compiled by Food Standards Australia New Zealand (FSANZ) into the AUSNUT 2011–2013 food recipe file [23]. The recipe file was used to derive the ingredients for each of the mixed dishes/composite foods reported in the survey. An example is homemade banana bread (group 1: flour, egg, cinnamon, banana; group 2: butter, sugar). A small number of dishes (*n* = 38) did not have a recipe in the database. In this situation, a recipe with comparable ingredients and nutrient composition was selected from the AUSNUT 2011–2013 database [23,24]. Ingredients in recipes were matched to the AUSNUT 2011–2012 nutrient composition database. Recipe weight change factors were used to calculate the nutrient composition of the recipe accounting for differences in macronutrients of the raw and cooked weight of the recipe.

Discretionary foods are defined by the ADG as food and beverages that do not fit into the five food group foods (i.e., vegetables; fruits; milk, cheese, yoghurts, and non-dairy alternatives; lean meat, poultry, fish, seafood, nuts, seeds, and legumes; grains and cereals) and are high in saturated fat, added sugars, added salt, and/or alcohol [6]. Discretionary foods were identified using the defined ABS discretionary food list [21]. Examples of foods classified under the two systems are shown in Table 1.

### 2.3. Dietary Assessment

Data on the foods and beverages consumed from midnight to midnight on the day preceding the interview were collected. The 24 h dietary recall used was the five-pass, Automated Multiple-Pass Method (AMPM) originally developed by the Agricultural Research Service of the United States Department of Agriculture (USDA). The tool was modified by FSANZ and the ABS to reflect the Australian food supply and provided the details for over 10,000 individual and combined foods. Interviewers used the Food Model Booklet developed to aid participants in estimating portions and included life-sized photographs of food and beverages and food and beverage containers to reflect those used within Australia. A computer-assisted telephone interview (CATI) was used to conduct a second 24 h recall for a proportion (63.6%) of the survey participants, but only the first day of the survey was used, as this is an accurate representation of the mean population nutrient intake. The AUSNUT 2011–2013 food nutrient database, compiled by FSANZ specifically for NNPAS, was used to derive food and beverages nutrient composition. The database contains the nutrient composition for 5740 foods and beverages reported in the survey and reflects the nutrient composition of the Australian food supply [24]. The Australian food supply and food preparation practices from 2011 to 2012 were, therefore, captured in AUSNUT 2011–2013 and the accompanying measures database.

### 2.4. Implausible Energy Reporting

To identify participants with implausible energy intake, the Goldberg equation was used. Participants with an energy intake to basal metabolic rate ratio (EI:BMR) of <0.87 were classified as low energy reporters (LER) and high energy reporters were defined as those with an EI:BMR > 2.75. An EI:BMR ratio ≥0.87 and ≤2.75 is within the 95% CI of plausible energy intake assuming a sedentary physical activity level of 1.55 [25]. Sensitivity analysis was conducted by repeating all analyses with and without LER to determine the effect of implausible energy intakes on the outcomes. As there were no differences in the direction of the point estimates, all analysis presented include the full adult sample.

### 2.5. Quantitative Variables

Energy intake was calculated per day with Atwater factors as total protein energy (17 kJ/gram) and non-protein energy (kJ) (i.e., fat × 37 + total sugars × 16 + starch × 17 + other available carbohydrates (dextrin + maltodextrin + raffinose + stachyose + other undifferentiated oligosaccharides + glycogen) × 17 + alcohol × 29 + sorbitol/mannitol/glycerol × 16 + maltitol × 13 + citric/malic/quinic acids × 10 + lactic/acetic acids × 15 + dietary fibre × 8 + resistant starch × 8 + polydextrose × 5).

The percentage energy (%E) from discretionary foods and the %E from UPF was calculated for each person and participants were categorised into tertiles and quintiles of discretionary food intake and UPF (%E) using PROC RANK (*n* = 3 and *n* = 5, respectively).

Weight was measured using digital scales and height was measured using a stadiometer. All physical measurements were voluntary, and women who had identified they were pregnant were not measured. Respondents were encouraged to remove their shoes and any heavy clothing, but this was voluntary [21].

### 2.6. Statistical Methods

Univariate and multivariate linear regression was used to determine the relationship between discretionary food (%E) and UPF (%E) and the intake of protein energy and non-protein energy. Multivariate linear regression was used to determine the relationship between BMI kg/m^2^ and discretionary food (%E) and UPF (%E) intake, classified into tertiles and quintiles. Multivariate analysis for energy intake was adjusted for the following factors: sex, age, smoking status. physical activity level, country of birth, and educational attainment. Complete case multivariate analysis for BMI was adjusted for sex, age, smoking status, and physical activity level. Estimates were weighted to reflect the Australian population distribution and probability of selection and replicate weights (the Jack-knife group delete one method) were used to compute standard errors. Analyses were conducted in SAS^®^ version 9.4: SAS Institute Inc. Significant differences were considered for *p* < 0·05.

## 3. Results

### 3.1. Participants

The demographics of participants by %E discretionary food intake and by UPF consumption is shown in Table 2. A total of 9341 adults reported dietary data on the first day of the survey and were included for analysis (Appendix A). Of these, 1486 adults were not included in the analysis of BMI due to missing data, as participants were only expected to record anthropometric data voluntarily, due to ethical considerations [21]. Participants who were older, female, higher socioeconomic index for area (SEIFA), university-educated, born in countries other than Australia or other English-speaking countries, and from major cities had lower mean intake from discretionary food (%E) and from UPF (%E) (Table 2).

### 3.2. Differences in UPFs and Discretionary Foods Classifications

In the AUSNUT 2011–2013 discretionary food list, 1631 foods of 5740 foods were classified as discretionary (28.4%). After disaggregation of the specified composite foods (e.g., homemade cakes) and mixed dishes (e.g., homemade pasta dishes) into individual ingredients, there were 2846 individual foods, of which 1016 (37.5%) were classed as discretionary foods and 1830 (64.3%) were five-food-group foods. Using the NOVA system to classify the disaggregated foods by degree of processing, 935 (32.9%) were classed as minimally processed foods, 60 (2.1%) were classed as culinary ingredients, 354 (12.4%) were classed as processed foods, and 1497 (52.6%) were classed as UPF.

All minimally processed foods, and most processed foods (83.9%) were classified as belonging to the five food groups under the ADG. A high proportion (63.3%) of culinary ingredients were classified as discretionary foods under the ADG. Similarly, a significant proportion of UPF (38.8%) were classified as belonging to the five food groups by the ADG (Figure 1a). The proportion of energy that came from discretionary foods and UPF, as classified by the ADG and NOVA, respectively, is shown in Figure 1b.

Across the surveyed population, discretionary foods accounted for 35.8 % and UPF accounted for 38.5% of total energy consumed. Almost 25% of daily energy came from foods classified as both discretionary food and UPF. However, there was a significant proportion of UPF classified as five-food-group foods, and 15.1% of daily energy came from five food group foods that were ultra-processed (Figure 1). Appendix A shows the proportion of energy from all foods reported in the National Nutrition and Physical Activity Survey classified by the NOVA classification system and the Australian Dietary Guidelines.

The observed agreement for participants to be classified in the same tertile of both discretionary food %E and UPF %E was 0.51, and the expected agreement was 0.3. A total of 38.4% of foods were classified in the adjacent tertile, and 10.4% were classified in the opposite tertile of discretionary food %E and UPF %E (i.e., the lowest discretionary food intake and the highest UPF intake and vice versa) (Table 3).

### 3.3. Energy Intakes

The energy intake from protein (MJ) and non-protein energy (MJ) for each tertile of discretionary food and UPF intake is shown in Figure 2. Participants in all tertiles of discretionary food intake and UPF intake consumed similar amounts of protein, with ~1.5 MJ of energy from protein for all groups. In contrast, energy from non-protein macronutrients increased between tertiles 1 and 3 for both UPF %E intake and discretionary food %E. The mean adjusted non-protein energy was 2.0 MJ between lowest and highest tertile for discretionary food %E and 0.6 MJ for UPF %E (Figure 2). The unadjusted mean non-protein energy intake difference between tertiles 1 and 3 %E UPF was 0.8 MJ (*p* < 0.001), whereas that for discretionary food %E was threefold higher (+2.2 MJ) (*p* < 0.001). The total mean adjusted energy intake, protein energy intake, and energy intake from macronutrients excluding energy from alcohol is shown in Table 4 for participants classified as quintiles in UPF and discretionary food (%E).

The difference in the unadjusted mean energy intake increased in a similar trend when participants were classified into quintiles, and the difference was 2.6 MJ between quintile 1 and quintile 5 between discretionary food, compared to 9.0 MJ between quintile 1 and quintile 5 for UPF. Excluding alcohol from the total energy intake reduced the difference in energy intake between quintile 1 and quintile 5 from 2.6 MJ to 1.6 MJ.

### 3.4. Body Mass Index and Intake of Discretionary Foods and UPF

In the unadjusted model, lower intakes of discretionary foods and UPF, categorised into tertiles by the proportion of energy from discretionary food and by the proportion of energy from UPF, were both significantly associated with a lower BMI (Table 5). In the multivariate model, participants classified into the lowest tertiles of discretionary food intake and UPF intake had the lowest BMIs by −1.0 and −1.1 kg/m^2^, respectively, compared to the highest consumers (Table 5). The magnitude in the changes was similar for UPF and DF.

## 4. Discussion

The present analysis of the dietary intake of a nationally representative sample of the Australian population demonstrates considerable overlap in the NOVA and ADG classification systems. The two systems classified all minimally processed foods and many processed foods as foods to be encouraged, and international dietary advice converges in this regard. However, there was some discrepancy between which system best identifies foods to be avoided in Australia, with many culinary ingredients being classified as unhealthy in ADG and most international dietary guidelines but not in NOVA. Likewise, our findings show that some foods identified as healthier by the ADG are classified as ultra-processed in the NOVA system. From the perspective of energy balance, both NOVA and ADG identified dietary patterns that elevate energy intake and were associated with overall higher BMI. The discretionary classification system was associated with both higher (quintile 5 = 8.6 MJ, Table 4) and lower (quintile 1, 5.9 MJ, Table 4) acute non-protein energy intake than the UPF classification (7.8 MJ vs. 6.9) and, consequently, a wider gap between highest and lowest quintiles (2.5 MJ vs. 0.6 MJ, respectively). Higher intakes of both UPF and discretionary food were associated with a higher BMI overall.

The NOVA and the ADG systems performed differently in their association with energy intakes, with high consumption of discretionary food related to larger increases in energy intake. At a fundamental level, weight gain occurs when energy intake exceeds energy expenditure and excess energy is stored [26]. Discrepancies as small as an additional 30 kJ per day may trigger weight gain, but an increase of 9 MJ has been estimated as needed to sustain the weight increases of the USA population [27,28]. After adjusting for relevant confounding factors, both classification systems were able to identify dietary patterns associated with higher energy intake and positive energy balance sufficient to drive weight gain.

It is important to examine the food groups in which the NOVA and ADG differ in terms of dietary guidance. Our analysis showed one point of difference to be centred on culinary ingredients. For example, the ADG advises against the use of added sugar and limits saturated fats unless energy requirements allow for some additional energy after nutrient requirements have been met [6]. The NOVA system, in contrast, limits added sugar when it is incorporated into UPF, but not consumer-added sugar, which is considered a culinary ingredient associated with home cooking [29]. However, in our analysis of the Australian food system, most of the culinary ingredients were used to make discretionary foods rather than healthier homemade dishes that can also be high in saturated fats, such as homemade or cafe/restaurant made cakes, biscuits, sweet and savoury pastries, and deep-fried vegetable products that may be detrimental to health if consumed in excess. A second difference between the systems is that a considerable proportion of five-food-group foods are classified as ultra-processed. Many of these that are low in saturated fat, e.g., ready-prepared-foods, may be harmful over time due to the impact of refined ingredients or exposure to harmful chemicals such as endocrine disruptors [2,30,31,32,33]. They may also have cumulative effects on energy storage over time if they contain highly refined, readily digestible sources of carbohydrates and fat relative to protein [34]. Additionally, alcohol, which is primarily classified as processed (rather than ultra-processed) in the NOVA system, also elevates acute energy intake, and sensitivity analysis demonstrated that some of the advantage held by the ADG system in predicting excess energy intake was due to the inclusion of alcohol in the discretionary category. The ADG and NOVA classification systems concur in identifying all minimally processed foods and many processed foods as healthy.

The mechanistic model of protein leverage for predicting the relationship between dietary composition and energy intake further explains the advantage of the ADG system compared to the UPF system in predicting high energy intakes in Australia. The protein leverage hypothesis posits that protein dilution of the food supply has been an important contributor to the obesity epidemic [16]. The comparatively smaller range of protein dilution in the NOVA system is due to the inclusion of relatively protein-dense foods as UPF. For example, flavoured yoghurt, classified as UPF but not discretionary food, has higher protein density. Conversely, we found that, in the Australian food system, many foods classified as discretionary food but not UPF have low protein density, e.g., homemade cakes. Additionally, low-protein diets that contain dietary fibre or resistant starch have been demonstrated to not lead to overconsumption [16,35]. Therefore, UPFs are unlikely to lead to higher energy intakes if they include unrefined sources of dietary fibre and resistant starch, such as commercial wholegrain bread or certain breakfast cereals with minimal added sugar.

It is important to note, however, that the close conformity between the patterns of macronutrient intakes observed in our analysis and predictions of the protein leverage hypothesis does not in itself provide definitive evidence of protein leverage. An alternative explanation could be that aggressive marketing, hyperpalatabilty, and other extrinsic factors drive excessive consumption of discretionary food and UPF, which both dilutes dietary protein and increases energy intake independent of protein appetite [32,36]. However, this hypothesis does not explain why absolute protein remained so constant; in contrast, constant protein intake is a central prediction of the protein leverage hypothesis. Furthermore, our interpretation is congruent with several other sources of evidence for protein leverage. These include recent advances in elucidating the biological mechanisms of protein appetites, demonstrating that fibroblast growth factor-21 (FGF-21) is the circulating metabolite when protein status is low in humans and rodents, acting on the brain to stimulate protein appetite [37,38]. Above all, protein leverage has been demonstrated in several RCTs in which diets were controlled for palatability, and aggressive marketing played no role [16,39,40]. It is, nonetheless, likely that, in ecological settings, protein leverage interacts with extrinsic factors. A two-stage model has been proposed for how this interaction might work [41], according to which hyper palatability, aggressive marketing, cheap price, and convenience associated with industrial foods attracts consumers to select and consume them. Due to their low protein content, this results in a reduction in the ratio of dietary protein to energy and, via protein leverage, energy over-consumption [20,34].

While our analysis implies that discretionary food consumption can explain the rising prevalence of obesity in Australia, these results may be country-specific or differ at an individual level. For example, an RCT examining the effect of UPF in the USA found similar increases in energy intake for discretionary food of 509 kcal (2118 kJ) greater compared to a control group consuming minimally processed foods [19]. Therefore, differences in the magnitude of effect could be due to difference in quality of the foods available in the Australian food supply or compared to those imposed on people in a laboratory setting, and the findings are likely to change dependent on overall diet quality and the relative fractions of UPF consumed. Analysis of NHANES dietary data derived a decline in %PE of 4.9% and an increase in energy intake of 0.9 MJ of non-protein energy between UPF quintiles, which is comparable to the increase in energy intake observed in the present study [20]. As discretionary foods led to greater protein dilution in Australia, this highlights the value of adopting an ecological model to examine the aetiology of the obesity epidemic.

A limitation of this study is that processing is not always adequately captured in the product description in the nutrient composition database, making the NOVA classification difficult to apply. Bread is one category where this is problematic as it is not possible to distinguish artisanal or homemade breads and mass-produced breads. To minimise differences, the NOVA coding system of Australian foods emulated that of previous research where foods were agreed upon by at least two researchers who applied expert knowledge in the Australian food supply, but there may still be some arbitrary misclassification of foods [22]. It should be noted that the Australian Bureau of Statistics discretionary food list was used to identify foods as discretionary according to the ADG. Although this list does not agree with the ADG in some aspects, as it was based on nutrient cut-points, it was used to more easily identify foods that were high in saturated fat, added sugars, and salt [42]. Misreporting dietary intake is a common limitation in dietary assessment [43], but sensitivity analysis revealed that this had a limited effect on the results presented here. Although the analysis was from a single day via a cross-sectional survey, the results are supported by other epidemiological studies and randomised controlled trials [3,44,45] and demonstrate the advantage of the discretionary food classification at a population level in the Australian food system. Our analysis used the most recent and comprehensive nutrition survey to have been conducted in Australia. Although this is 10 years old, we do not see reason to expect that factors driving the main conclusions would have changed over this period. Indeed, with the continued rise in industrial foods and obesity in Australia [46], we suspect that they might be even more relevant at present.

There is increased financial burden for individuals and communities amounting to 20.5 billion AUD annually in direct health costs in Australia due to poor diets [47]. The prevalent solution in Australia and other developed countries has been a focus on individual responsibility through setting-based approaches or public health information campaigns. Without further policy, legislative, and structural changes to the food environment, efforts to prevent noncommunicable disease will be frustrated [48]. There have been calls to level the playing field by matching the heavy marketing of UPF/discretionary food with government spending on advertising of healthful dietary patterns [49]. Even for some of the worst aspects of the food environment such as vending machines, there is good evidence that simply providing customers with healthier choices is enough to improve the purchases made [50]. Efforts to reduce the intake of foods that dilute protein energy have the potential to contribute to and strengthen obesity prevention.

## 5. Conclusions

UPFs have been associated with the rise in obesity in many countries, including Australia, but our study highlights the importance of monitoring dietary patterns with ecological studies to determine the efficacy of different food classification systems in different food environments. While there was considerable overlap in both the ADG and the NOVA classification systems, the apparent advantage of the discretionary classification system in Australia is that it detects a greater spread in the extent to which problem foods dilute dietary protein and, hence, their effect on energy intake via protein leverage. Dietary guidance targeted at reducing intake of discretionary foods in Australia may thus have greater potential for supporting obesity prevention than a focus on UPF. Excessive energy intake is not, however, the only aspect of health for which dietary guidelines are relevant. Further studies are needed to determine how different classification systems relate to other associations between diet and all aspects of health in Australia and other countries.

## Figures and Tables

**Figure 1 nutrients-14-03942-f001:**
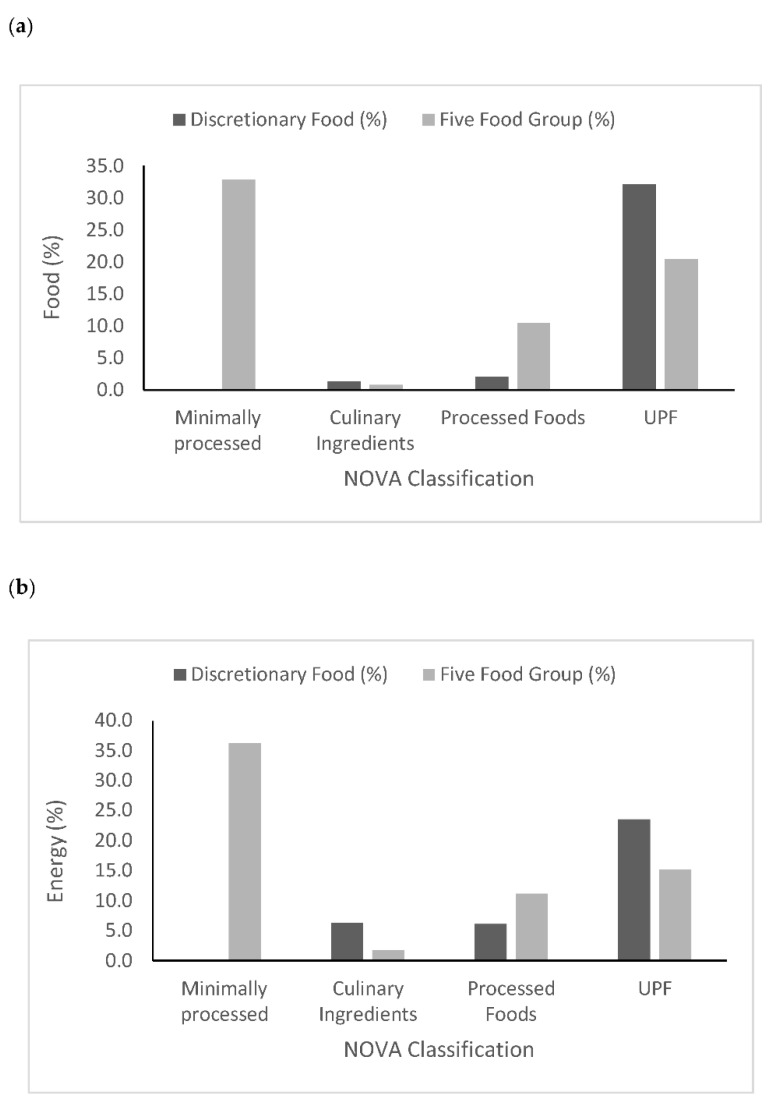
(**a**) Total number of foods reported in the survey (*n* = 2486); (**b**) proportion of daily energy from foods classified as discretionary foods or five-food-group foods (according to the Australian Dietary Guidelines) and by degree of processing (according to the NOVA classification system i.e., minimally processed, culinary ingredients, processed foods, or ultra-processed foods (UPF)) as reported by adults in the National Nutrition and Physical Activity Survey (*n* = 9431).

**Figure 2 nutrients-14-03942-f002:**
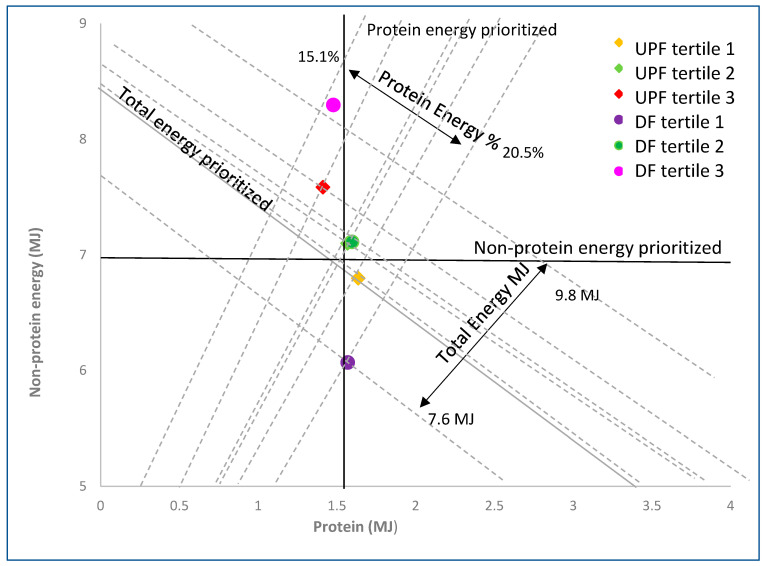
Mean protein and non-protein energy intake for participants categorised into tertiles of discretionary food (DF) and ultra-processed food (UPF). The positively sloped radials indicate the proportion of energy from protein from total energy intake and demonstrate protein dilution with increased intake of discretionary and UPF. The negatively sloped diagonals indicate total daily energy intake. The data points line up along the solid vertical line demonstrating that protein energy intake is prioritised. If total daily energy intake is prioritised, the values line up along the solid negative radial, while the horizontal line indicates the situation if non-protein energy is prioritised.

**Table 1 nutrients-14-03942-t001:** Contribution daily energy intake (%) of selected foods by NOVA classification system and Australian Dietary Guidelines.

Classification System	Five Food Groups Foods	Discretionary Foods	Disaggregated Discretionary Foods
Minimally processed foods	Tea, coffee, home squeezed juice, water, barley, cornmeal, millet, oats, quinoa, sago, rice bran, rice, wheat germ, wheat bran, couscous, flour, semolina, tapioca, noodles, pasta, natural muesli, fish, seafood, apple, pear and all frozen, fresh, and dried fruit, nuts, eggs, beef, lamb, pork, veal, goat, chicken, turkey, milk, plain yoghurt, seeds, psyllium, potato, carrot and all fresh, frozen and dried vegetables, herbs, lentils, beans		Homemade and takeaway foods † including sweet and savoury pastry, cakes, pies, French toast, cakes, muffins, slices, puddings, tarts, spring rolls, pizzas, waffles, deep fried fish and vegetables, cream-based desserts, sauces, jams and icings, pizzas (pepperoni, ham and cheese, meat lovers), quiche
Culinary ingredients	Olive oil, vinegar, flaxseed oil, rice bran oil, yeast, gelatine, canola oil, soybean oil, peanut oil, sunflower oil, vegetable oil, canola oil, gelatine, baking powder	Cream, butter, lard, ghee, sour cream, sugar, honey, and salt	
Processed foods	Homemade and artisan breads, salted nuts, nut spreads, cheese, tinned fruit, tinned meat and seafood, peanut butter, tinned vegetables, and legumes	Bacon, wine, beer, cider, chutneys, pickles, condensed milk, jam	
Ultra- processed foods	Commercial fruit juice, beverage (milo), commercial breads, commercial English muffins, instant noodles, breakfast cereals with low/no added sugar, savoury biscuits, commercial scones, fast food pizzas (<5 g saturated fat), fast food burgers (<5 g saturated fat), frozen meals, tinned spaghetti, commercial crumpets, margarine, sausages (<5 g saturated fats), breaded chicken, flavoured yoghurts, processed cheese, flavoured milks, soymilk, oat milk, tofu, tempeh, canned and packet soups (lower sodium), baked beans, intense sweeteners, oral supplements	Fruit drinks, sweetened drinks, cordial, soft drinks, flavoured beverage bases, sweet buns, breakfast cereals, commercial sweet biscuits, commercial garlic bread, ice cream cones, wafer commercial cakes, muffins, slices, pastries, commercial savoury pastries, fast food burgers, pizzas, frozen meals including pizzas, donuts, butter blends, Copha, frozen fish, sausages, ham, salami, other processed meats, chicken nuggets, ice cream, dairy desserts, packet soups, gravies, marinades, sauces, dressings and dips, fast foods and frozen potato fries, savoury snack foods, chocolates, confectionary, muesli bars, spirits, protein powder, yeast spreads	

† Discretionary mixed dishes that were disaggregated into their component ingredients.

**Table 2 nutrients-14-03942-t002:** Percentage energy for discretionary food (DF) and ultra-processed foods (UPF) by demographics for Australian adults (*n* = 9341).

Demographics	%	(SE)	Mean DF %E		*p*-Trend	Mean UPF %E		*p*-Trend
Age								
19–30 years	23.1	0.3	34.0	(0.7)		43.9	(0.8)	
31–50 years	37.4	0.3	32.9	(0.4)		38.0	(0.4)	
51–70 years	28.7	0.2	31.2	(0.4)		34.5	(0.5)	
71+ years	10.8	0.1	31.6	(0.7)	0.0057	36.5	(0.7	<0.0001
Gender								
Female	49.4	(0.1)	30.7	(0.4)		37.5	(0.4)	
Male	50.6	(0.1)	34.3	(0.4)	<0.0001	38.8	(0.5)	0.0473
SEIFA								
Lowest (quintile 1)	18.1	(1.0)	33.8	(0.7)		40.1	(0.8)	
Middle (quintile 2–3)	59.7	(1.4)	32.6	(0.3)		38.4	(0.4)	
Highest (quintile 5)	22.2	(1.0)	31.2	(0.6)	0.0184	35.9	(0.7)	0.0013
Educational Attainment								
No tertiary education	38.8	(0.6)	33.5	(0.5)		40.1	(0.8)	
Vocational education	35.5	(0.7)	34.2	(0.4)		38.4	(0.4)	
University education	25.7	(0.7)	28.6	(0.5)	<0.0001	35.9	(0.7)	0.0013
Country of Birth								
Australia	68.8	(0.9)	34.6	(0.3)		40.3	(0.4)	
Other English-speaking countries	11.6	(0.4)	34.1	(0.8)		37.6	(0.8)	
Other	19.6	(0.8)	24.1	(0.6)	<0.0001	31.0	(0.7)	<0.0001
Geographic Area								
Major cities	71.5	(0.6)	31.3	(0.3)		37.3	(0.3)	
Inner regional	19.1	(0.8)	36.1	(0.6)		40.7	(0.7)	
Other	9.4	(0.8)	34.3	(0.9)	<0.0001	39.3	(1.1)	<0.0001
Energy Reporting Status								
Low (EI:BMR ≤ 0.87)	16.8	(0.5)	25.5	(0.7)		36.9	(0.7)	
Plausible (EI:BMR > 0.87)	69.2	(0.7)	34.6	(0.3)		38.7	(0.4)	
Missing	14.0	(0.4)	30.8	(0.7)	<0.0001	37.1	(0.7)	0.0427

SEIFA, socio-economic index for area.

**Table 3 nutrients-14-03942-t003:** Proportion of participants that were classified in the same, adjacent, and opposite tertile for percentage energy (%E) of discretionary food and ultra-processed food (UPF) (%E).

Discretionary Food (%E)	UPF (%E)	(*n*)	%
Tertile 1—lowest	Tertile 1—lowest	1721	18.4
Tertile 1—lowest	Tertile 2—middle	930	10.0
Tertile 1—lowest	Tertile 3—highest	462	4.9
Tertile 2—middle	Tertile 1—lowest	884	9.5
Tertile 2—middle	Tertile 2—middle	1319	14.1
Tertile 2—middle	Tertile 3—highest	911	9.8
Tertile 3—highest	Tertile 1—lowest	508	5.4
Tertile 3—highest	Tertile 2—middle	865	9.3
Tertile 3—highest	Tertile 3—highest	1741	18.6

Proportions in same tertile (51.2%), adjacent tertile (38.4%), and opposite tertile (10.4%). Kappa weights: observed agreement, 0.51; chance-expected agreement, 0.33.

**Table 4 nutrients-14-03942-t004:** Total energy, protein energy and energy excluding alcohol for participants classified by proportion of energy from discretionary food (DF) and ultra-processed foods (UPF).

	DF (%E)	UPF (%E)	DF (%E)		UPF (%E)		DF (No Alcohol) (%E)	UPF
	Total Energy (MJ)	Total Energy (MJ)	Total Energy (MJ)	Protein (MJ)	Total Energy (MJ)	Protein (MJ)	P + C + F (MJ)	P + C + F (MJ)
Quintile	Mean	Mean	Adj. Mean †	Adj. Mean †	Adj. Mean †	Adj. Mean †	Adj. Mean †	Adj. Mean †
1	7.4	8.2	7.5	1.6	8.5	1.6	7.4	7.8
2	8.3	8.7	8.4	1.6	8.9	1.6	8.0	8.4
3	8.7	8.6	8.8	1.6	8.8	1.5	8.4	8.3
4	9.1	8.8	9.1	1.5	8.8	1.4	8.5	8.5
5	10.0 ***	9.1 ***	10.0 ***	1.4 ***	9.1 **	1.3 ***	9.0 ***	8.7 ***

† Adjusted (adj.) for age, sex, physical activity level, smoking status, educational attainment, and country of birth. P, protein; C, carbohydrate; F, fat. ** Significant linear trend across quintiles = 0.001; *** significant linear trend across quintiles < 0.0001.

**Table 5 nutrients-14-03942-t005:** Change in body mass index (BMI) with changes in intake of discretionary food as defined by the Australian Dietary Guidelines (ADG) and ultra-processed food (UPF) as a proportion of energy as defined by the NOVA classification system.

Food Intake (Range)		Model 1: Change in BMI (SE)		*p*-Value	Model 2: Change in BMI (SE)		*p*-Value
ADG classification							
DF—tertile 1	(0.0–≤21.8)	−0.8	(0.2)		−1.0	(0.2)	
DF—tertile 2	(>21.8–41.6)	−0.2	(0.2)		−0.4	(0.2)	
DF—tertile 3	(≥41.7–100)	Ref		0.0003	Ref		<0.0001
NOVA classification							
UPF—tertile 1	(≥0.0–<29.4)	−0.9	(0.2)		−1.1	(0.2)	
UPF—tertile 2	(≥29.4–<49.7)	−0.4	(0.2)		−0.5	(0.2)	
UPF—tertile 3	(≥49.7–100.0)	Ref		0.0003	Ref		<0.0001

%E, percentage energy. *p*-Values for trend were determined with linear and multiple liner regression. Model 1: univariate model. Model 2: adjusted for sex, age, smoking status (current smoker, daily; current smoker, <weekly; current smoker, at least once per week but not daily; ex-smoker, never smoked); physical activity level (sedentary (very low); sedentary (no exercise); not stated; low; moderate; high); energy intake: basal metabolic rate ratio. Survey weights were applied.

## Data Availability

Data are available from the Australian Bureau of Statistics upon request https://www.abs.gov.au/ (accessed on 30 January 2021).

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
