# Peer review of "A Comparison of the Australian Dietary Guidelines to the NOVA Classification System in Classifying Foods to Predict Energy Intakes and Body Mass Index"

_nutrients, 2022, doi:10.3390/nu14193942_

Round 1

Reviewer 1 Report

1.       Overall, I believe this to be a well-designed study. However, I do not believe the findings have wide applicability. While there may be a need for researchers to understand differences in dietary measurement tools, this study compares the tools and examines intake of discretionary foods, UPF, and protein, but it does not suggest whether one tool is superior or whether there are specific environments in which this may be the case. While both survey tools showed that lower protein intake was associated with discretionary food and UPF intake, this relationship is not a novel finding. The authors need to spell out more specifically the novelty, relevance, and applicability of the study.

2.       Line 16: grammatical error “specifically those high saturated fat”. Please correct and clarify so that the sentence makes sense.

3.       Lines 70-75: I had to read this sentence three times to understand it. I suggest re-writing this sentence to make it more readable.

4.       Lines 132-146: Is there a reason this paragraph is italicized?

5.       Lines 193-203: Can you please explain further why a sedentary physical activity level of 1.55 was chosen and what this value indicates? Does this refer to the EI:BMR ratio?

6.       Line 428: subject-verb error

7.       Discussion: The discussion section is overly long. I suggest decreasing it by about 20%.

8.       There are minor punctuation errors throughout. For example, there is a comma on line 83 that should not be there, as it is placed between the subject and verb with no intervening phrase set off by commas. I suggest a careful check of the document, as these errors negatively affect the overall readability of the manuscript.

Author Response

Overall, I believe this to be a well-designed study. However, I do not believe the findings have wide applicability. While there may be a need for researchers to understand differences in dietary measurement tools, this study compares the tools and examines intake of discretionary foods, UPF, and protein, but it does not suggest whether one tool is superior or whether there are specific environments in which this may be the case. While both survey tools showed that lower protein intake was associated with discretionary food and UPF intake, this relationship is not a novel finding. The authors need to spell out more specifically the novelty, relevance, and applicability of the study.

The novelty and significance of our paper is twofold:

  1. With the shift from nutrient-based to food-based dietary guidelines, it is more important than ever to provide simple and succinct guidance for consumers at to which foods to avoid. Dietary classification systems are the main tool for this, and it is therefore of considerable importance to examine how well they perform the task for which they are designed. Our study showed that the system currently in use in Australia performs better in terms of direction for avoiding excess energy intake, which is among the highest of public health priorities in this country, than does the more prominent NOVA classification.
  2. Our ecological analysis of foods, combined with our test for protein leverage, explains why the Australian system performs better than the NOVA system within Australia. The reason is that it is more sensitive to low-protein foods that engage with protein leverage than is the NOVA system.
  • The study suggests that particular dietary classificatory systems might perform differently in different food environments, and highlights the importance of examining their performance using ecological analyses

We have made the necessary changes in the paper to emphasise this, including a substantial editing of the abstract.

  1. Line 16: grammatical error “specifically those high saturated fat”. Please correct and clarify so that the sentence makes sense.

Thank you for the correction, we have edited the abstract

  1. Lines 70-75: I had to read this sentence three times to understand it. I suggest re-writing this sentence to make it more readable.

We have re-worded this sentence at lines 69-74

  1. Lines 132-146: Is there a reason this paragraph is italicized?

No, thank you we have corrected this

  1. Lines 193-203: Can you please explain further why a sedentary physical activity level of 1.55 was chosen and what this value indicates? Does this refer to the EI:BMR ratio?

Line 208: A reference (Black, 2008) with the validation study has been included

  1. Line 428: subject-verb error

No change required (now line 445)

  1. Discussion: The discussion section is overly long. I suggest decreasing it by about 20%.

We have shortened the discussion by removing a paragraph

  1. There are minor punctuation errors throughout. For example, there is a comma on line 83 that should not be there, as it is placed between the subject and verb with no intervening phrase set off by commas. I suggest a careful check of the document, as these errors negatively affect the overall readability of the manuscript.

Readability has been checked, thank you

Reviewer 2 Report

Manuscript 1913113:  A comparison of the Australian Dietary Guidelines to the NOVA classification system in classifying foods to predict energy intakes and body mass index.

Overall evaluation

The work is interesting and highlights some limitations of the NOVA food classification. In particular, those relating to bread or fortified foods. In my opinion it needs to have some minor corrections.

Major remarks

In table 1 the first age group ranges from 19 to 50 years. Is it possible to divide it into smaller classes? This is because the dietary patterns of young people are different from those of middle-aged subjects.

Minor remarks

Lines 132-146                        Why these sentences are in italic ?

Lines 149-150                        There is probably an extra bar space between “The recipes for” and  “2,585 mixed ….”

Line 257                                 What does SEIFA stand for ?

Figure 1 a) and b)                It would be better to use more different colors for bars. Furthermore, instead of Five Food Group, it would be better use “NOVA classification” or “NOVA”

Author Response

Manuscript 1913113:  A comparison of the Australian Dietary Guidelines to the NOVA classification system in classifying foods to predict energy intakes and body mass index.

 Overall evaluation

The work is interesting and highlights some limitations of the NOVA food classification. In particular, those relating to bread or fortified foods. In my opinion it needs to have some minor corrections.

Thank you for your feedback

Major remarks

In table 1 the first age group ranges from 19 to 50 years. Is it possible to divide it into smaller classes? This is because the dietary patterns of young people are different from those of middle-aged subjects.

We have include the age range from 19-30 in Table 1

Minor remarks

Lines 132-146                        Why these sentences are in italic ?

 In error, thank you

Lines 149-150                        There is probably an extra bar space between “The recipes for” and  “2,585 mixed ….”

 Corrected, thank you

Line 257                                 What does SEIFA stand for ?

Socio-economic index for area

Figure 1 a) and b)                It would be better to use more different colors for bars. Furthermore, instead of Five Food Group, it would be better use “NOVA classification” or “NOVA”

Thank you, we have increased the contrast of figure 1

Round 2

Reviewer 1 Report

I appreciate the revisions made by the authors to improve this paper. In particular, the new emphasis on the importance of the study has raised my evaluation.